# Subarachnoid Hemorrhage in Hospitalized Renal Transplant Recipients with Autosomal Dominant Polycystic Kidney Disease: A Nationwide Analysis

**DOI:** 10.3390/jcm8040524

**Published:** 2019-04-17

**Authors:** Wisit Cheungpasitporn, Charat Thongprayoon, Patompong Ungprasert, Karn Wijarnpreecha, Wisit Kaewput, Napat Leeaphorn, Tarun Bathini, Fouad T. Chebib, Paul T. Kröner

**Affiliations:** 1Division of Nephrology, Department of Medicine, University of Mississippi Medical Center, Jackson, MS 39216, USA; 2Division of Nephrology and Hypertension, Mayo Clinic, Rochester, MN 55905, USA; charat.thongprayoon@gmail.com (C.T.); chebib.fouad@mayo.edu (F.T.C.); 3Clinical Epidemiology Unit, Department of Research and Development, Faculty of Medicine, Siriraj Hospital, Mahidol University, Bangkok 10700, Thailand; p.ungprasert@gmail.com; 4Department of Medicine, Division of Gastroenterology and Hepatology, Mayo Clinic, Jacksonville, FL 32224, USA; karnjuve10@gmail.com (K.W.); thomaskroner@gmail.com (P.T.K.); 5Department of Military and Community Medicine, Phramongkutklao College of Medicine, Bangkok 10400, Thailand; wisitnephro@gmail.com; 6Department of Nephrology, Department of Medicine, Saint Luke’s Health System, Kansas City, MO 64111, USA; napat.leeaphorn@gmail.com; 7Department of Internal Medicine, University of Arizona, Tucson, AZ 85721, USA; tarunjacobb@gmail.com

**Keywords:** autosomal dominant polycystic kidney disease, epidemiology, hospitalization, kidney transplantation, subarachnoid hemorrhage

## Abstract

Background: This study aimed to evaluate the hospitalization rates for subarachnoid hemorrhage (SAH) among renal transplant patients with adult polycystic kidney disease (ADPKD) and its outcomes, when compared to non-ADPKD renal transplant patients. Methods: The 2005–2014 National Inpatient Sample databases were used to identify all hospitalized renal transplant patients. The inpatient prevalence of SAH as a discharge diagnosis between ADPKD and non-ADPKD renal transplant patients was compared. Among SAH patients, the in-hospital mortality, use of aneurysm clipping, hospital length of stay, total hospitalization cost and charges between ADPKD and non-ADPKD patients were compared, adjusting for potential confounders. Results: The inpatient prevalence of SAH in ADPKD was 3.8/1000 admissions, compared to 0.9/1000 admissions in non-ADPKD patients (*p* < 0.01). Of 833 renal transplant patients with a diagnosis of SAH, 30 had ADPKD. Five (17%) ADPKD renal patients with SAH died in hospitals compared to 188 (23.4%) non-ADPKD renal patients (*p* = 0.70). In adjusted analysis, there was no statistically significant difference in mortality, use of aneurysm clipping, hospital length of stay, or total hospitalization costs and charges between ADPKD and non-ADPKD patients with SAH. Conclusion: Renal transplant patients with ADPKD had a 4-fold higher inpatient prevalence of SAH than those without ADPKD. Further studies are needed to compare the incidence of overall admissions in ADPKD and non-ADPKD patients. When renal transplant patients developed SAH, inpatient mortality rates were high regardless of ADPKD status. The outcomes, as well as resource utilization, were comparable between the two groups.

## 1. Introduction

Subarachnoid hemorrhage (SAH) is a major clinical problem worldwide, associated with a poor prognosis, long-term morbidity, and extremely high mortality [1,2,3]. In the United States, over 30,000 cases of SAH occur annually [4,5]. Although global SAH has decreased each year from 1960 through 2017 [6], approximately 30% to 50% of patients who developed SAH died at 60 days. Furthermore, over 30% of survivors subsequently suffered neurologic deficits post-SAH [1,2,3,4,7,8,9]. Major causes of SAH include ruptured intracranial aneurysm, cerebral arteriovenous malformation, and traumatic brain injury [6,10].

Among patients with autosomal dominant polycystic kidney disease (ADPKD), a disorder that affects the kidneys and other organs caused by mutations in PKD1 and PKD2, a wide spectrum of vascular abnormalities have been described including intracranial aneurysms (and dolichoectasias), thoracic aorta and cervicocephalic artery dissections, and coronary artery aneurysms [11,12]. Compared to the general population, the prevalence of intracranial aneurysms in ADPKD patients is approximately five times higher and is estimated at 4% to 22.5% [13,14,15,16,17,18,19]. Among ADPKD patients with a family history of SAH/intracranial aneurysms, the frequency is three to five times higher than the general population [12]. Thus, screening is recommended for intracranial aneurysms among ADPKD patients with (1) family or past medical history of intracranial aneurysm presence or rupture; (2) symptoms suggesting intracranial aneurysm; (3) occupations in which loss of consciousness may be fatal; (4) upcoming major elective surgery; and (5) patient subjective concern for possible intracranial aneurysm presence [20,21,22].

Despite the recommendation for intracranial aneurysm screening prior to major elective surgery, such screening for ADPKD patients is not consistently performed among all transplant centers in the USA during pre-transplant assessment for renal transplantation [23,24]. In addition, previously published studies on the increased risk of SAH after renal transplantation in patients with ADPKD have not yielded relevant information, or have been underpowered [25,26,27,28,29,30,31].

Thus, we conducted this study using a nationwide inpatient USA database to evaluate the hospitalization rates for SAH among renal transplant patients with ADPKD and its outcomes, when compared to non-ADPKD renal transplant patients.

## 2. Methods

### 2.1. Data Source

The 2005–2014 National Inpatient Sample (NIS) databases were used to conduct this retrospective cohort study. The NIS is the largest inpatient all-payer database that is publicly available in the US. This database was developed by the Agency for Healthcare Research and Quality (AHRQ) as part of its Healthcare Cost and Utilization Project (HCUP). The dataset for the studied years contains more than 78 million hospitalizations, which in itself is a 20% stratified sample of over 4000 non-federal acute care hospitals in 44 states of the United States, and is representative of 95% of hospitalizations nationwide. This dataset included codes for principal diagnosis, secondary diagnoses, and codes for procedures performed during the hospitalization.

### 2.2. Study Population

Initially, renal transplant patients were identified using the International Classification of Diseases, Ninth Revision, Clinical Modification (ICD-9-CM) codes of v42.0. The ADPKD status in this cohort was identified using the ICD-9-CM code 753.13. The associated diagnosis of SAH was identified using the ICD-9-CM code 430. Patients undergoing elective hospital admission or patients undergoing renal transplantation during the same admission were excluded from the analysis.

### 2.3. Variable Definition

Patient characteristics included age, gender, ethnicity, median income in patients’ zip code, family history of stroke, and insurance type. Hospital characteristics included hospital region, teaching status, number of hospital beds, urban location, and weekend admission. The Healthcare Cost and Utilization Project (HCUP) divides the US into four census regions based on geographical location: Northeast, Midwest, South and West. The vital status at the end of hospitalization, length of hospital stay (LOS), and total hospitalization charges were abstracted from the database. To account for patient comorbidities, the Deyo adaptation of the Charlson Comorbidity Index was used, which was appropriate for the large database analysis [32].

### 2.4. Outcomes

The outcome for primary analysis was to determine the inpatient prevalence of SAH as a discharge diagnosis in renal transplant patients with ADPKD, compared to renal transplant patients without ADPKD. The outcomes for secondary analysis were to compare in-hospital mortality, the use of aneurysm clipping, length of hospital stay, and expenditures between ADPKD and non-ADPKD patients with SAH. Expenditures included total hospitalization charges and hospitalization costs. Total hospitalization charges represented the amount of financial resources that each hospital billed for providing its service for each patient, whereas hospitalization costs represented the amount of money spent by each hospital in providing the patient care. Hospitalization costs were calculated by multiplying the cost-to-charge ratios for the respective hospitals with the total hospitalization charges. Cost-to-charge ratios were provided by the Healthcare Cost and Utilization Project (HCUP) for each hospitalization in the database in order to enable this calculation. Since this study used datasets for 10 different calendar years, costs and charges were adjusted for inflation using the consumer price index and converting them to 2014 $USD equivalents.

### 2.5. Statistical Analysis

Discharge-level weights on the HCUP nationwide databases were used to estimate the total number of renal transplant patients that had an associated diagnosis of SAH. Descriptive statistics were used to identify the patient characteristics. Fisher’s exact test was used to compare proportions. Students’ *t*-test was used to compare means. A hybrid multivariate logistic regression model was built by first conducting a univariate regression analysis on variables that were identified from other studies as being relevant to the outcome. If these variables impacted the outcome in any direction with a *p*-value of <0.01, they were included in the multivariate logistic regression model. In multivariate logistic regression, odds ratios and means were adjusted for age, gender, insurance type, family history of stroke, the median income in patients’ zip code, hospital region, urban location, number of hospital beds and teaching status. All statistical analyses were performed using STATA, Version 13 (StataCorp LP, College Station, TX, USA).

## 3. Results

### 3.1. Inpatient Prevalence of SAH as a Discharge Diagnosis

Out of 382,516,561 patients admitted to hospitals during the study period, 918,478 were identified as having had a history of renal transplant. ADPKD patients who underwent renal transplant had higher inpatient prevalence of SAH as a discharge diagnosis than non-ADPKD renal transplant patients (3.8 vs. 0.9 cases per 1000 discharges; *p* < 0.01).

### 3.2. Patient and Hospital Characteristics in SAH Patients

In total, 833 patients had an associated diagnosis of SAH. These included 30 ADPKD renal transplant patients and 803 non-ADPKD renal transplant patients. The ADPKD renal transplant patients were older, had higher comorbidity burden, and were more likely to be admitted to teaching hospitals than non-ADPKD renal patients. There was no significant difference in terms of weekend admission, the median income in patients’ zip code, insurance type, hospital region, urban location, and number of hospital beds between the two cohorts (Table 1).

### 3.3. Mortality

A total of 5 (17%) ADPKD renal transplant patients with SAH died in hospitals compared to 188 (23.4%) non-ADPKD renal transplant patients (*p* = 0.70). Similarly, in adjusted analysis, there was no difference in in-hospital mortality between the two groups, with an adjusted odds ratio (aOR) of 0.87 (95% confidence interval (CI) 0.12–6.23; *p* = 0.89) (Table 2).

### 3.4. Use of Aneurysm Clipping

Ten (33.0%) ADPKD renal transplant patients with SAH underwent aneurysm clipping, compared to 137 (17.1%) non-ADPKD renal transplant patients (*p* = 0.32). On adjusted analysis, the use of aneurysm clipping in ADPKD renal transplant patients was not significantly higher than non-ADPKD renal transplant patients (aOR: 2.02; 95% CI 0.28–14.81; *p* = 0.49) (Table 2).

### 3.5. Hospital Length of Stay

The mean LOS in ADPKD renal transplant patients with SAH was 9.8 days, compared to 8.9 days in the non-ADPKD renal transplant SAH patients. Although the mean additional LOS in ADPKD renal transplant patients with SAH was 3.0 days shorter than non-ADPKD renal transplant patients with SAH, this was not statistically significant (95% CI: −10.1–4.1, *p* = 0.41) on adjusted analysis (Table 3).

### 3.6. Total Hospitalization Costs and Charges

The mean hospital cost for ADPKD renal transplant patients with SAH was $ 30,519, while the mean hospital cost for non-ADPKD renal transplant patients with SAH was $ 33,526 (*p* = 0.04). Although the ADPKD renal transplant patients with SAH had a mean hospital cost $ 1086 lower than the non-ADPKD renal transplant patients, the difference was not statistically significant (95% CI: −22,548–20,376, *p* = 0.92) on adjusted analysis (Table 3).

The mean total hospitalization charge for ADPKD renal transplant patients with SAH was $ 85,682, while the mean total hospitalization charges for non-ADPKD renal transplant patients with SAH was $ 112,514. Although ADPKD renal transplant patients with SAH had a mean hospitalization charge $ 14,944 lower than the non-ADPKD renal transplant patients, this was not statistically significant (95% CI: −73,293–43,404, *p* = 0.62) on adjusted analysis (Table 3).

## 4. Discussion

In this study utilizing the USA Nationwide Inpatient Sample database, we demonstrated that renal transplant patients with ADPKD had a 4-fold higher inpatient prevalence of SAH than those without ADPKD. When renal transplant patients developed SAH, the inpatient mortality rate was high (around 20 to 30%), regardless of ADPKD status. In addition, the use of aneurysm clipping for SAH and hospital LOS were comparable among renal transplant patients with and without ADPKD.

In the previous analysis using USA Renal Data System registry data, Lentine et al. reported a decreased risk of SAH in renal transplant patients when compared to end-stage renal disease (ESRD) patients on the transplant waiting list [33]. Additionally, among ESRD patients on dialysis, it has been shown that ADPKD is associated with an increased risk of SAH [34,35]. Despite an overall reduction in SAH risk after renal transplantation [33], our study demonstrates a higher relative frequency of SAH among ADPKD patients compared to those without ADPKD and aims to raise awareness that SAH remains an important concern in the post-transplantation population.

The risk of intracranial aneurysms among ADPKD patients increased with age [36,37] and its prevalence is substantially increased after 45 years of age, especially among ADPKD Caucasian patients [38]. The average age of patients with ADPKD and ESRD is greater than 45 years [34,35,38]. However, intracranial aneurysm screening during pre-transplant evaluation for renal transplantation was a requirement for some but not all transplant centers in the USA, despite recommendation for intracranial aneurysm screening prior to major elective surgery [23,24]. Furthermore, a very recent study by Flahault et al. including 495 ADPKD patients suggested that systematic screening was cost-effective and provided a gain of 0.68 quality-adjusted life years compared to targeted screening in only those with a familial history of intracranial aneurysms [21]. The investigators concluded that intracranial aneurysm screening could be proposed to all ADPKD patients regardless of family history of stroke [21].

In our study involving renal transplanted patients, after adjusting for potential confounders including family history of stroke, the relative frequency of SAH among ADPKD patients remained significantly higher compared to those without ADPKD. This may suggest potential development of SAH in renal transplant patients with ADPKD, regardless of family history of stroke. When patients develop SAH, the mortality rate is high regardless of renal transplant status [27,33,39,40,41]. In our study, nearly 25% of renal transplant patients with SAH died during hospital admission. After adjusting for potential confounders, we found no differences in in-hospital mortality, rate of aneurysm clipping, hospital LOS, hospital costs and total hospitalization charges among ADPKD and non-ADPKD renal transplant patients with SAH.

Several limitations of this study must be acknowledged. Firstly, although the utilization of the NIS database enables an assessment of inpatient relative frequency and burden of SAH in renal transplant patients in the USA, potential inaccuracies in ICD-9-CM coding are significant limitations to our study. Secondly, the data relating to types of mutation among ADPKD patients were limited in this study. Since patients with *PKD2*-associated ADPKD usually develop ESRD at an older age compared to renal-transplanted ADPKD patients (mean age: 79.7 vs. 58.9 years) [22,42,43], it is likely that our study represents the outcomes of SAH among renal transplant patients with *PKD1*-associated ADPKD. Thirdly, this is an analysis of an inpatient USA database and, thus, it does not consider the broader USA outpatient renal transplant population nor the renal transplant population in other countries. Because of the nature of the NIS database, our study included only inpatient renal transplant patients and might be subject to selection bias. Without knowing the total number of all ADPKD and non-ADPKD renal transplant patients, the overall admission rates for any reasons cannot be calculated. Fourthly, although the findings of our study highlight the burden of SAH in ADPKD renal transplant patients, it cannot be concluded that screening all ADPKD patients prior to renal transplantation is cost-effective. Nevertheless, physicians should take into account the risk of SAH among ADPKD renal transplant patients. Furthermore, the practice of intracranial aneurysm screening in ADPKD patients might vary between hospitals. However, information regarding the practice of intracranial aneurysm screening was not available in our study. Lastly, given the administrative nature of the dataset, it was not possible to investigate the effects of medication, such as immunosuppressants, on economic burden and mortality among ADPKD and non-ADPKD renal transplant patients with SAH.

## 5. Conclusions

In conclusion, in this study using the USA Nationwide Inpatient Sample database, we demonstrate higher inpatient relative frequency of SAH among ADPKD renal transplant patients when compared with non-ADPKD renal transplant patients. In-hospital mortality due to SAH among renal transplant patients was high, regardless of ADPKD status. The use of aneurysm clipping for SAH and hospital length of stay were comparable between ADPKD renal transplant patients and those without ADPKD.

## Figures and Tables

**Table 1 jcm-08-00524-t001:** Patient and hospital characteristics.

Patient Characteristics	Non-ADPKD (*n* = 803)	ADPKD (*n* = 30)	*p*-Value
Mean age (years)	53.4	58.9	<0.01
Female gender (%)	491 (61.2%)	18 (59.1%)	0.7
Ethnicity			
Caucasian	522 (65%)	21 (70%)	
African American	112 (14%)	3 (10%)	0.02
Hispanic	145 (18%)	5 (18%)	
Other	24 (3%)	1 (2%)	
Weekend admission	193 (24%)	5 (18%)	0.17
Income in zip code			
$1–37,999	217 (27%)	6 (21%)	
$38,000–47,999	208 (26%)	8 (28%)	0.36
$48,000–63,999	193 (24%)	7 (22%)	
>$64,000	185 (23%)	9 (29%)	
Insurance			
Medicare	313 (39%)	9 (29%)	
Medicaid	104 (13%)	4 (13%)	0.28
Private	321 (40%)	14 (48%)	
Self-Pay	64 (8%)	3 (10%)	
Charlson score			
0	0 (0%)	0 (0%)	
1–2	586 (73%)	13 (42%)	<0.01
>3	214 (27%)	17 (58%)	
Hospital Region			
Northeast	145 (18%)	6 (19%)	
Midwest	177 (22%)	7 (23%)	0.15
South	297 (37%)	8 (25%)	
West	185 (23%)	10 (33%)	
Urban Location	779 (97%)	29 (98%)	0.39
Hospital Number of Beds			
Small	48 (6%)	2 (6%)	
Medium	145 (18%)	4 (14%)	0.64
Large	610 (76%)	24 (80%)	
Hospital Teaching Status	602 (75%)	26 (88%)	<0.01

ADPKD, adult polycystic kidney disease.

**Table 2 jcm-08-00524-t002:** Outcomes and procedures of ADPKD and non-ADPKD patients with SAH.

**Outcome**	**Adjusted OR**	**95% CI**	***p*-Value**
Mortality	0.87	(0.12–6.23)	0.89
**Procedure**	**Adjusted OR**	**95% CI**	***p*-Value**
Aneurysm clipping	2.02	0.28–14.81	0.49

**Table 3 jcm-08-00524-t003:** Hospital length of stay and expenditure differences between ADPKD and non-ADPKD patients with SAH.

Additional Expenditures	Mean Difference	95% CI	*p*-Value
Hospital length of stay (days)	−3.0	−10.1–4.1	0.41
Total hospitalization cost	−$1086	−$22,548–$20,376	0.92
Total hospitalization charge	−$14,944	−$73,293–$43,404	0.62

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
