# Peer review of "Subarachnoid Hemorrhage in Hospitalized Renal Transplant Recipients with Autosomal Dominant Polycystic Kidney Disease: A Nationwide Analysis"

_jcm, 2019, doi:10.3390/jcm8040524_

Reviewer 1 Report

In this study, Wisit Cheungpasitporn et al. report that, utilizing the U.S. Nationwide Inpatient Sample database, renal transplant patients with ADPKD had 4-fold higher inpatient prevalence of SAH than those without ADPKD. When renal transplant patients developed SAH, inpatient mortality rate was nearly 20 to 30%, regardless of ADPKD status. In addition, the uses of aneurysm clipping for SAH and hospital length of stay were comparable among renal transplant patients with and without ADPKD. This is a well-designed, well-described study but important information appears missing.

[Major Points]

1) The authors reveal higher INPATIENT frequency of SAH among ADPKD renal transplant recipients compared to non-ADPKD recipients. It is not investigated whether overall admission rates are similar between ADPKD and non-ADPKD recipients. This is a major limitation of this work and should be described in the abstract.

2) Did the admitted kidney transplantation recipients and the patients who developed SAH belong to centers which require intracranial aneurysm screening during pre-transplant evaluation? Did they actually undergo the screening?

Author Response

Reviewer #1
In this study, Wisit Cheungpasitporn et al. report that, utilizing the U.S. Nationwide Inpatient Sample database, renal transplant patients with ADPKD had 4-fold higher inpatient prevalence of SAH than those without ADPKD. When renal transplant patients developed SAH, inpatient mortality rate was nearly 20 to 30%, regardless of ADPKD status. In addition, the uses of aneurysm clipping for SAH and hospital length of stay were comparable among renal transplant patients with and without ADPKD. This is a well-designed, well-described study but important information appears missing

Response: We thank you for reviewing our manuscript and for your critical evaluation. We really appreciated your input and found your suggestions very helpful.

 Comment #1

The authors reveal higher INPATIENT frequency of SAH among ADPKD renal transplant recipients compared to non-ADPKD recipients. It is not investigated whether overall admission rates are similar between ADPKD and non-ADPKD recipients. This is a major limitation of this work and should be described in the abstract.

Response: The reviewer is very thorough and has made very good point. We appreciated the reviewer input. Because the NIS database included only inpatient admission, we cannot compare the overall admission rates for any reasons between ADPKD and non-ADPKD renal transplant recipients. The following statements have been added in the limitation section.  

Thirdly, this is an analysis of an inpatient U.S. database and, thus limits the generalizability to the broader U.S. outpatient renal transplant population, or to the renal transplant population in other countries. Because of the nature of NIS database, our study included only inpatient renal transplant patients and might be subjected to the selection bias. Without knowing the total number of all ADPKD and non-ADPKD renal transplant patients, the overall admission rates for any reasons cannot be calculated.

We respected the reviewer and we have also additionally emphasized that our study is based on data on U.S. hospitalized patients in the abstract.

 Comment #2

Did the admitted kidney transplantation recipients and the patients who developed SAH belong to centers which require intracranial aneurysm screening during pre-transplant evaluation? Did they actually undergo the screening?

Response: The reviewer raised very important point. The NIS database unfortunately did not contain information regarding the practice of intracranial aneurysm screening in ADPKD patients. The following statements have been added in the limitation section

The practice of intracranial aneurysm screening in ADPKD patients might vary between hospitals. However, the information regarding the practice of intracranial aneurysm screening was not available in our study.

All authors thank the Editors and reviewers for their valuable suggestions. The manuscript has been improved considerably by the suggested revisions! 

Reviewer 2 Report

Cheungpasitporn et al. evaluate a very upsetting question that has raised increasing concern during the last years; the importance of SAH after transplantation in ADPKD patients. Even if it’s known that is a cause of mortality and morbidity in ADPKD kidney recipients, experts do not agree on the need for aneurysm screening before transplantation. In fact this study is very useful to raise awareness of this complication once again. Nonetheless, there is a very important selection bias which doesn’t allow to draw general conclusions.

MAJOR COMMENTS

The main limitation of the study is the selection bias entailed by only including in-patient renal transplant recipients. Moreover, the authors also exclude those KT recipients with SAH diagnosis undergoing kidney transplantation during the same admission. Thus, although few, they may have skipped cases of SAH in the immediate post-transplant period.

Another major concern is the fact that the authors used ICD-9 CM coding in order to include kidney transplant recipients in the study. Nonetheless, they should specify if they checked the graft status when the SAH occurred. At some point one can think patients may be coded as v42.0. according to ICD-9 CM classification even if the graft is not functioning any more. It would be desirable to know the time from the kidney transplant to the SAH event.  

On the other hand, in the introduction the authors state that global SAH incidence has decreased over the past decade. The period taken into account in the study was 2005-2014 which represents only half of the past decade. 

MINOR COMMENTS

-          Page 3. Line 114, 115. The sentence should be re-written

-          Page 3. Line 119. P-value should be re-written as < 0.01

-          Table 1.  Specify the number of patients (n) per condition, not only percentages.

-          Table 1. Define what is a small, medium, large hospital.

-          Page 4. Line 139. The sentence should be re-written A total of 5 (17%) ADPKD renal transplant patients or renal transplant recipients.

-          Table 2. It is included in the paragraph 3.3 Mortality. Additional expenditures and Hospital Length of stay should be written in a separate table.

Author Response

Response to Reviewer #2

 Cheungpasitporn et al. evaluate a very upsetting question that has raised increasing concern during the last years; the importance of SAH after transplantation in ADPKD patients. Even if it’s known that is a cause of mortality and morbidity in ADPKD kidney recipients, experts do not agree on the need for aneurysm screening before transplantation. In fact this study is very useful to raise awareness of this complication once again. Nonetheless, there is a very important selection bias which doesn’t allow to draw general conclusions.

 Response: We thank you for reviewing our manuscript and for your critical evaluation. We really appreciated your input and found your suggestions very helpful.

Comment #1

The main limitation of the study is the selection bias entailed by only including in-patient renal transplant recipients. 

Response: The reviewer raised very important point. We appreciated the reviewer input. We agree with the reviewer and we have added the following statements in the limitation section as the reviewer’s suggestion.  

Thirdly, this is an analysis of an inpatient U.S. database and, thus limits the generalizability to the broader U.S. outpatient renal transplant population, or to the renal transplant population in other countries. Because of the nature of NIS database, our study included only inpatient renal transplant patients and might be subjected to the selection bias.

Comment #2

Moreoverthe authors also exclude those KT recipients with SAH diagnosis undergoing kidney transplantation during the same admission. Thus, although few, they may have skipped cases of SAH in the immediate post-transplant period.

Response: The reviewer raised very important point. We respected the reviewer. Thus, we checked our data and found that no kidney transplant patients had subarachnoid hemorrhage diagnosis during the same admission as kidney transplant procedure in NIS database.

Comment #3

Another major concern is the fact that the authors used ICD-9 CM coding in order to include kidney transplant recipients in the study. Nonetheless, they should specify if they checked the graft status when the SAH occurred. At some point one can think patients may be coded as v42.0. according to ICD-9 CM classification even if the graft is not functioning any more. It would be desirable to know the time from the kidney transplant to the SAH event.

Response: We appreciated the reviewer’s thorough review. We queried the code for renal replacement therapy during the subarachnoid hemorrhage-related hospitalizations to determine whether kidney allograft was functioning. We included only kidney transplant patients with no dialysis during hospitalization in the analysis. 

Comment #4

On the other hand, in the introduction the authors state that global SAH incidence has decreased over the past decade. The period taken into account in the study was 2005-2014 which represents only half of the past decade.

Response: We greatly appreciated the reviewer’s comment. The data on National Inpatient Sample databases were limited between the year of 2005 and 2014. The reviewer raised important point on the introduction and we have revised the introduction to truly represent the year of study from the cited study in the introduction “global SAH has decreased each year from 1960 through 2017”

Comment #5

Page 3. Line 114, 115. The sentence should be re-written

Response: We agree with the reviewer. We have clarified “HCUP” as Healthcare Cost and Utilization Project (HCUP) and have re-written the sentence as the reviewer’s suggestion.

Comment #6

Page 3. Line 119. P-value should be re-written as < 0.01

Response: We have changed P-value < 0.01 as the reviewer’s suggestion

Comment #7

Table 1.  Specify the number of patients (n) per condition, not only percentages.

Response: We agree with the reviewer. The number of counts has been added in table 1 as the reviewer’s suggestion.

Patient Characteristics

No ADPKD (n=803)

ADPKD (n=30)

p-value

Mean age   (years)

53.4

58.9

<0.01< span="">

Female   gender (%)

491 (61.2%)

18 (59.1%)

0.70

Ethnicity
       Caucasian
       African American
       Hispanic
       Other

  522 (65%)
  112 (14%)
  145 (18%)
  24 (3%)

  21 (70%)
  3 (10%)
  5 (18%)
  1 (2%)

0.02

Weekend   admission

               193   (24%)

5 (18%)

0.17

Income   in zip code

     $1 - $37,999

     $38K – 47,999

     $48K – 63,999

     > $64,000

217 (27%)

208 (26%)

193 (24%)

185 (23%)

6 (21%)

8 (28%)

7 (22%)

9 (29%)

0.36

Insurance
       Medicare
       Medicaid
       Private
       Self-Pay

  313 (39%)
  104 (13%)
  321 (40%)
  64 (8%)

  9 (29%)
  4 (13%)
  14 (48%)
  3 (10%)

  0.28

Charlson   score

     0

     1 – 2

     > 3

0 (0%)

586 (73%)

214 (27%)

0 (0%)

13 (42%)

17 (58%)

<0.01< span="">

Hospital   Region

     Northeast

     Midwest

     South

     West

145 (18%)

177 (22%)

297 (37%)

185 (23%)

6 (19%)

7 (23%)

8 (25%)

10 (33%)

0.15

Urban   Location

779 (97%)

29 (98%)

0.39

Hospital   Number of Beds

     Small

     Medium

     Large

48 (6%)

145 (18%)

610 (76%)

2 (6%)

4 (14%)

24 (80%)

0.64

Hospital   Teaching Status

602 (75%)

26 (88%)

<0.01< span="">

 Comment #8

Table 1. Define what is a small, medium, large hospital.

Response: We agree with the reviewer. The definition of hospital size was according to Agency for HealthCare Research and Quality (AHRQ) and depended on the region, location, and teaching status of the hospitals. We have now clarified this in data source section.

BEDSIZE CATEGORIES

Location and Teaching Status

Hospital Bedsize

Small

Medium

Large

NORTHEAST REGION

Rural

1-49

50-99

100+

Urban,   nonteaching

1-124

125-199

200+

Urban,   teaching

1-249

250-424

425+

MIDWEST REGION

Rural

1-29

30-49

50+

Urban,   nonteaching

1-74

75-174

175+

Urban,   teaching

1-249

250-374

375+

SOUTHERN REGION

Rural

1-39

40-74

75+

Urban,   nonteaching

1-99

100-199

200+

Urban,   teaching

1-249

250-449

450+

WESTERN REGION

Rural

1-24

25-44

45+

Urban,   nonteaching

1-99

100-174

175+

Urban,   teaching

1-199

200-324

325+

 Comment #9

Page 4. Line 139. The sentence should be re-written A total of 5 (17%) ADPKD renal transplant patients or renal transplant recipients

Response: We appreciated the reviewer’s thorough review comment. We have made this change as the reviewer’s suggestion.

Comment #10

Table 2. It is included in the paragraph 3.3 Mortality. Additional Expenditures and Hospital Length of stay should be written in a separate table.

Response: We agree with the reviewer. We have now separated the table of mortality and aneurysm clipping (Table 2) from the table of hospital length of stay and expenditure, as the reviewer’s suggestion.

All authors thank the Editors and reviewers for their valuable suggestions. The manuscript has been improved considerably by the suggested revisions! 

Round  2

Reviewer 1 Report

Incidence of overall admissions in ADPKD and non-ADPKD patients was not available in this study. This information should be included in the ABSTRACT. Otherwise, the conclusion is not supported by the findings provided. If word limitation is tight, the last sentence of the abstract can be omitted.

Author Response

Point-by-point response to reviewers’ comments:
Reviewer #1

Incidence of overall admissions in ADPKD and non-ADPKD patients was not available in this study. This information should be included in the ABSTRACT. Otherwise, the conclusion is not supported by the findings provided. If word limitation is tight, the last sentence of the abstract can be omitted.

Response: We once again thank you for reviewing our manuscript and for your critical evaluation. We really appreciated your input and found your suggestions very helpful.

We agree with the reviewer’s important point. As the reviewer’s suggestion, we have added “Further study to compare the incidence of overall admissions in ADPKD and non-ADPKD patients was needed.” in the conclusion. In addition, we also included “Hospitalized” in the new title “Subarachnoid Hemorrhage in Hospitalized Renal Transplant Recipients with Autosomal Dominant Polycystic Kidney Disease: A Nationwide Analysis” to truly represent patient population in our study.
All authors thank the Editors and reviewers for their valuable suggestions. The manuscript has been improved considerably by the suggested revisions! 

Reviewer 2 Report

Dear authors,

As commented previously, conclusions can not be drawn from this study for all the APDK transplant patients, as the evaluation only takes into consideration in-patients recipients. By remarking on this fact, the manuscript would be suitable for publication. Nonetheless it would be desirable to state in the title the population the study focus on, in order to prevent misconceptions.

MAJOR COMMENTS

It would be desirable to know whether all the grafts were functioning at the time of SAH and the time from renal transplant to SAH episode.

Author Response

Point-by-point response to reviewers’ comments:
Reviewer #2
Comment 1

As commented previously, conclusions cannot be drawn from this study for all the APDK transplant patients, as the evaluation only takes into consideration in-patients recipients. By remarking on this fact, the manuscript would be suitable for publication. Nonetheless it would be desirable to state in the title the population the study focus on, in order to prevent misconceptions.

Response: We once again thank you for reviewing our manuscript and for your critical evaluation. We really appreciated your input and found your suggestions very helpful.

We agree with the reviewer’s important point. We have added “Hospitalized” in the new title “Subarachnoid Hemorrhage in Hospitalized Renal Transplant Recipients with Autosomal Dominant Polycystic Kidney Disease: A Nationwide Analysis” to truly represent patient population in our study and to prevent misconceptions, as the reviewer’s suggestion.

Comment 2

It would be desirable to know whether all the grafts were functioning at the time of SAH and the time from renal transplant to SAH episode.

Response: We appreciated the reviewer’s thorough review. We agree with the reviewer and we additionally queried the code for renal replacement therapy during the subarachnoid hemorrhage-related hospitalizations to determine whether kidney allograft was functioning. In this study, we included only kidney transplant patients with functioning grafts (with no dialysis during hospitalization) in the analysis

All authors thank the Editors and reviewers for their valuable suggestions. The manuscript has been improved considerably by the suggested revisions! 
